# Corrosion Compatibility of Stainless Steels and Nickel in Pyrolysis Biomass-Derived Oil at Elevated Storage Temperatures

**Jiheon Jun** [1], **Yi-Feng Su** [1], **James R. Keiser** [1], **John E. Wade IV** [1], **Michael D. Kass** [2], **Jack R. Ferrell III** [3], **Earl Christensen** [4], **Mariefel V. Olarte** [5] and **Dino Sulejmanovic** [1,*]

1. Materials Science and Technology Division, Oak Ridge National Laboratory, Oak Ridge, TN 37831, USA
2. Buildings and Transportation Science Division, Oak Ridge National Laboratory, Oak Ridge, TN 37830, USA
3. National Renewable Energy Laboratory, Golden, CO 80401, USA
4. Alder Fuels, Golden, CO 80401, USA
5. Pacific Northwest National Laboratory, Richland, WA 99354, USA
* Correspondence: sulejmanovid@ornl.gov

**Abstract:** Corrosion compatibility of stainless steels and nickel (Ni200) was assessed in fast pyrolysis bio-oil produced from pyrolysis of high ash and high moisture forest residue biomass. Sample mass change, ICP-MS and post-exposure electron microscopy characterization was used to investigate the extent of corrosion. Among the tested samples, type 430F and type 316 stainless steels (SS430F and SS316) and Ni200 (~98.5% Ni) showed minimal mass changes (less than 2 mg·cm$^{-2}$) after the bio-oil exposures at 50 and 80 °C for up to 168 h. SS304 was also considered to be compatible in the bio-oil due to its relatively low mass change (1.6 mg·cm$^{-2}$ or lower). SS410 samples showed greater mass loss values even after exposures at a relatively low temperature of 35 °C. Fe/Cr values from ICP-MS data implied that Cr enrichment in stainless steels would result in a protective oxide layer associated with corrosion resistance against the bio-oil. Post exposure characterization showed continuous and uniform Cr distribution in the surface oxide layer of SS430F, which showed a minimal mass change, but no oxide layer on a SS430 sample, which exhibited a significant mass loss.

**Keywords:** biomass pyrolysis; bio-oil; stainless steel; corrosion compatibility; Nickel

## 1. Introduction

Biomass-derived pyrolysis oils have been considered as renewable fuel sources which can contribute to reduction of $CO_2$ emissions [1–8]. Among the developed processes to derive bio crude oils from biomass feedstocks, fast pyrolysis (FP) of forest residues (FR) with varying ash and moisture contents was extensively studied at the National Renewable Energy Laboratory (NREL), Golden, CO, USA. NREL produced several of these bio-oils, referred to as FR1, FR2, FR3 and FR4 [9]. Similar to other FP bio-oils [10–17], these FR bio-oils are acidic as they contain low to high molecular weight organic acids [9], which is a specific corrosion concern for carbon steel and low-alloy steel-based fuel storage tanks [18–20]. The corrosion of several metallic alloys by organic acids was investigated in the past [21–29]. However, the corrosion of alloys in bio-oil environments is a topic that has not been widely studied, and has been identified as a challenging problem for materials scientists due to the chemical complexity of bio-oil liquids from the difference in biomass feedstocks, the presence of impurities such as oxygen and moisture, and processing conditions [9,13,14,30–35].

To investigate the structural material compatibility in FR bio-oils, corrosion mass changes of several ferrous alloys were analyzed after sample exposure in the bio-oils at 50 °C up to 1000 h [36–38], which revealed that type 304 L and 316 L stainless steels (SS304L and SS316L) are almost immune to corrosion, type 409 stainless (SS409) is marginally resistant,

but carbon and low alloy steels are highly susceptible to corrosion. This indicates that the contents of Cr and Ni added for corrosion resistance played a crucial role in mitigating or preventing the alloy mass loss. Additionally, a previous electrochemical corrosion evaluation of Cr-alloyed steels and stainless steels concluded that a critical Cr content for corrosion resistance would be 12~13 wt.% for FR3 bio-oil exposure at room temperature [38].

To further study the alloy corrosion behavior in a pyrolysis bio-oil, stainless steels with different Cr and Ni contents and commercial purity Ni (Ni200) were exposed to FR bio-oil at several different temperatures (35, 50 and 80 °C) and for different times (48, 96 and 168 h). Different forms of alloy samples, including coupon, mesh and wire, were obtained to investigate whether corrosion performance is affected by the production process history. In addition, inductively coupled plasma mass spectroscopy (ICP-MS) measurements were also performed to quantify the amount of metal cations leached into the bio-oil during the exposure tests. Microscopic characterization was then conducted for selected post-exposure alloy samples to reveal any microstructural features associated with corrosion attack by FR bio-oil.

## 2. Materials and Methods

### 2.1. Materials

Several metallic alloys, in wrought rod and plate as well as extruded wire and mesh forms, were purchased from different commercial vendors, and their chemical compositions, dimensions and densities are summarized in Tables 1 and 2. For composition analysis, inductively coupled plasma and combustion techniques were utilized. Note that there are two SS410 mesh types, (1) and (2), with lower carbon contents (≤0.04 wt.%) and different wire radius values (see Table 2). Additionally, note that SS430F is similar to SS430 but contains a high sulfur content to increase machinability. The obtained rod and plate stocks were machined to 16 mm diameter disk samples with a 1 mm diameter wide round hole, and then the flat and side surfaces of the samples were finished with 600 grit SiC paper. An example of a SS410 disk sample is shown in Figure 1a. Mesh and wire samples were rinsed with ethanol and cleaned in an ultrasonic water bath without any mechanical surface finishing. To calculate the surface area, the diameter and thickness of each disk sample were measured, and the measured weight, wire radius and alloy density were used to calculate the estimated surface areas of wire and mesh samples where the cross-sectional round areas were ignored due to their insignificance. To measure the initial sample mass, an analytical balance (Mettler Toledo model XP205) with an accuracy of ~±0.04 mg was used.

**Table 1.** Chemical compositions of metallic alloys in weight percentage by inductively coupled plasma mass spectrometry (ICP-MS) and combustion techniques.

| | **Fe** | **Cr** | **Ni** | **Mo** | **Mn** | **Cu** | **Si** | **C** | **N** | **O** | **S** | **Others** |
|---|---|---|---|---|---|---|---|---|---|---|---|---|
| (A) SS410 Coupon * | 88.6 | 11.9 | 0.23 | 0.05 | 0.47 | 0.16 | 0.3 | 0.13 | 0.012 | 0.0026 | <0.001 | 0.06 V 0.02 Co |
| SS410 Mesh (1) | Bal. | 13.2 | 0.22 | 0.03 | 0.42 | 0.09 | 0.32 | 0.03 | 0.07 | 0.01 | 0.001 | 0.06 V 0.02 Co |
| SS410 Mesh (2) | Bal. | 13.1 | 0.2 | 0.02 | 0.46 | 0.05 | 0.31 | 0.04 | 0.063 | 0.0069 | 0.001 | 0.04 V |
| SS410 Wire ** | Bal. | 12.5 | - | - | 1 | - | 1 | <0.15 | - | - | <0.03 | |

**Table 1.** *Cont.*

| | Fe | Cr | Ni | Mo | Mn | Cu | Si | C | N | O | S | Others |
|---|---|---|---|---|---|---|---|---|---|---|---|---|
| (A) SS430 Coupon * | 81.1 | 16.9 | 0.15 | 0.05 | 0.26 | 0.13 | 0.33 | 0.035 | - | - | 0.002 | 0.14 Ti 0.08 V 0.02 Co |
| SS430 Mesh | Bal. *** | 17.3 | 0.19 | <0.01 | 0.29 | 0.03 | 0.39 | 0.04 | 0.03 | 0.01 | 0.03 | 0.05 Co 0.02 V |
| SS430 Wire ** | Bal. | | | | | | | | | | | |
| (B) SS430F Coupon * | 80.4 | 17.5 | 0.3 | 0.33 | 0.43 | 0.17 | 0.32 | 0.032 | 0.042 | 0.009 | 0.326 | 0.06 V |
| SS304 Mesh and wire ** | Bal. | 18 | 9 | - | 2 | - | 0.75 | <0.08 | 0.1 | - | <0.03 | |
| SS316 Mesh and wire ** | Bal. | 17 | 11 | 2.2 | 1 | - | 0.5 | <0.07 | - | - | <0.02 | |
| Ni200 Mesh and wire ** | <0.4 | - | Bal. | - | <0.35 | <0.25 | <0.35 | <0.15 | - | - | <0.01 | |
| (B) Ni200 Coupon ** | | | | | | | | | | | | |

* Previously reported in [39,40], ** Nominal composition, *** Approximately 81.7 wt.%. Originally from: (A) wrought plate and (B) wrought rod.

**Table 2.** Radius of mesh and wire alloy samples and approximate densities of the alloys.

| | SS410 Mesh (1) | SS410 Mesh (2) | SS430 Mesh | SS304 Mesh | SS316 Mesh | Ni200 Mesh | SS410 Wire | SS430 Wire | SS304 Wire | SS316 Wire | Ni200 Wire |
|---|---|---|---|---|---|---|---|---|---|---|---|
| Wire radius (cm) | 0.03175 | 0.0179 | 0.03175 | 0.02032 | 0.02032 | 0.00635 | 0.0406 | 0.0406 | 0.0406 | 0.0406 | 0.0315 |
| Density (mg·cm$^{-3}$) | 7750 | 7750 | 7720 | 8000 | 8000 | 8900 | 7750 | 7720 | 8000 | 8000 | 8900 |

FR3 bio-oil, obtained from NREL, was used for corrosion exposure tests. This bio-oil was produced by fast pyrolysis liquefaction, resulting in high moisture (~26 wt.%) bio-oils [9]. Further details of the fast pyrolysis process, as well as chemical and physical properties of FR3 bio-oil, can be found in the previous literature [9,41]. FR3 bio-oil tends to separate into low viscosity phase and high viscosity dense sludge after extended storage at room temperature (22~23 °C). To minimize this phase separation, the FR3 bio-oil was thoroughly agitated prior to its use for corrosion exposure tests.

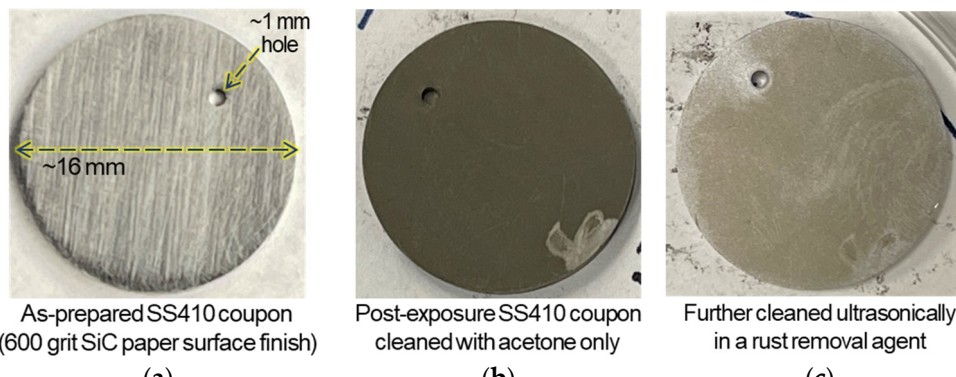

As-prepared SS410 coupon (600 grit SiC paper surface finish) **(a)** | Post-exposure SS410 coupon cleaned with acetone only **(b)** | Further cleaned ultrasonically in a rust removal agent **(c)**

**Figure 1.** Appearances of SS410 coupon ($\varnothing \approx 16$ mm) (**a**) in as-prepared condition. After exposure to FR3 bio-oil at 50 °C for 48 h, the same coupon received (**b**) acetone-cleaning and then (**c**) further cleaning to remove corrosion products as much as possible.

### 2.2. Bio-Oil Exposure

For corrosion exposure tests, each alloy specimen, suspended by a polytetrafluoroethylene (PTFE) wire, was immersed in FR3 bio-oil contained in a glass vial. The mass of FR3 bio-oil was approximately 25 g per vial for sufficient specimen mass to FR3 bio-oil mass ratio, as summarized along with the surface areas in Table 3. Note that the variation of specimen to bio-oil mass ratio, among coupons, meshes and wires, was considered as one of the factors which result in corrosion mass change. The exposure tests were conducted at three different temperatures, 35, 50 and 80 °C, and for three different durations, 48, 96 and 168 h.

**Table 3.** Specimen mass to bio-oil mass ratio for different types of alloy samples. The ranges of specimen surface areas are also presented for each sample type.

|  | Coupon Type | Mesh Type | Wire Type |
|---|---|---|---|
| $\dfrac{\text{FR3 bio oil mass}}{\text{specimen mass}}$ | 5< and <16 | 65< and <270 | 60< and <320 |
| Specimen surface area/cm$^2$ | 4.45~5.44 | 2.19~10.8 | 0.57~2.37 |

After the tests, the exposed specimens were rinsed with acetone and methanol to remove bio-oil residue and then ultrasonically cleaned in a mineral-oil based rust remover solution for 5 min or longer. Figure 1b,c show the appearance of an exposed coupon after acetone/methanol cleaning and additional rust remover cleaning, respectively. Pictures of the exposed wire and mesh samples after cleaning are presented in Figure A1 in the Appendix A. Upon the completion of cleaning, post-exposure sample mass was measured using the same analytical balance to calculate the mass changes in mg·cm$^{-2}$ units.

### 2.3. ICP-MS and Microscopic Characterization

After the exposure and removal of alloy samples, the remaining bio-oils in glass vials were sent to an external analytical chemistry lab for ICP-MS to quantify the key metal elements leached from the exposed alloys. An acid digestion method, described in detail in Table A1 in the Appendix A, was applied for the bio-oil samples, and then the digests were diluted by 2% nitric acid solution prior to ICP-MS measurements. For dilution, the volume ratio of digest and 2% nitric acid solution was 1:5. The instrument used for ICP-MS was a Thermo Finnigan Element XR model.

Some of the post-exposure alloy samples were cross-sectionally analyzed to investigate microstructural features from corrosion by FR3 bio-oil. Alloy samples for characterization were first cross-sectioned, mounted in epoxy. Then, the epoxy-mounted cross sections were polished using SiC papers and diamond suspension solutions for a mirror polish surface. A scanning electron microscope (SEM, TESCAN, Warrendale, PA, USA) equipped with

energy dispersive X-ray spectrometry (EDS, Oxford Instruments, Abingdon, UK) was used for microscopic characterization and chemical analysis of the cross-sectioned alloy samples.

## 3. Results and Discussion

### 3.1. Mass Change and ICP Data

Mass changes of SS410, SS430 and SS430F in coupon, mesh or wire form, which were exposed in FR3 bio-oil for 48 h, are compared in Figure 2. SS430F coupon exhibited insignificant mass changes at both 50 and 80 °C exposures. SS430, on the other hand, showed mass change values varying considerably between coupon, mesh and wire forms as noted in the following cases—high mass loss of coupon but low mass change of wire at 80 °C exposure; relatively high mass loss of mesh but low mass changes of coupon and wire at 50 °C exposure. These results imply that SS430 can suffer corrosion in FR3 bio-oil due to some variations, possibly from thermal and mechanical process histories, as well as subtle difference in chemical compositions. Unlike SS430, SS410 commonly showed relatively low and high mass loss values at 35 and 50 °C exposures, respectively, in all specimen types. At 80 °C exposure, SS410 coupon and wire had significant mass losses greater than $-30$ mg·cm$^{-2}$. Based on the results, SS410 is expected to suffer fast corrosion attack in FR3 bio-oil at 50 °C or higher temperatures.

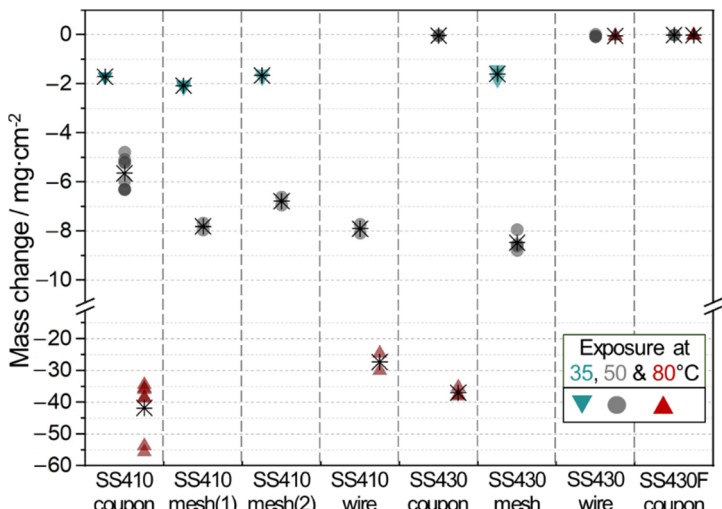

**Figure 2.** Mass change data of SS410, SS430 and SS430F samples. Downward triangles, circles and upward triangles are individual mass change data at 35, 50 and 80 °C exposures, respectively. Asterisk symbols are the average values. Note that there was an excluded outlier, $-0.017$ mg·cm$^{-2}$ from SS410 mesh (2) after exposure at 80 °C for 48 h. SEM and EDS characterization for the sample is provided in Figure A2 in the Appendix A.

Mass change values of SS304, SS316 and Ni200 in mesh and wire forms are plotted in Figure 3 for 48 h bio-oil exposure. Most data at all temperatures are less than $-0.5$ mg·cm$^{-2}$, but SS304 wire exposed at 80 °C shows mass loss close to $-1.6$ mg·cm$^{-2}$. The difference in mass loss between SS304 mesh and wire could be attributed to alloy process history and subtle changes in chemical compositions. Overall, SS304 would not suffer significantly from bio-oil corrosion as the greatest mass loss data (from wire form at 80 °C) were still lower than the mass loss of SS410 at 35 °C exposure (close to $-2$ mg·cm$^{-2}$). SS316 and Ni200 samples, based on their small mass change data, are expected to be highly resistant to corrosion in FR3 bio-oil. Excellent corrosion resistances of SS304 and SS316 were previously reported in different bio-oil exposures at 50 °C for extended durations up to 1000 h [36–38]. However, a previous report stated that a Ni-based alloy would be corrosion-susceptible in bio-oils [42], which contradicts the corrosion results in this work.

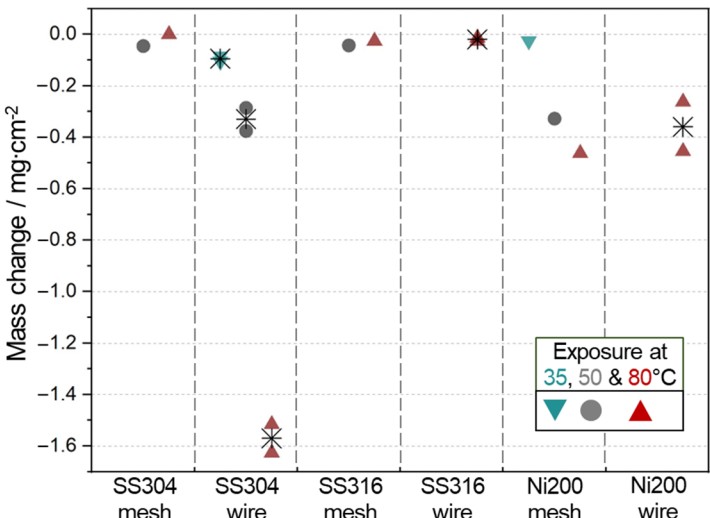

**Figure 3.** Mass change data of SS304, SS316 and Ni200 samples. Downward triangles, circles and upward triangles are individual mass change data at 35, 50 and 80 °C exposures, respectively. Asterisk symbols are the average values.

The concentration of (Fe + Cr) cations in bio-oils, analyzed by ICP-MS, are presented for SS410, SS430 and SS430F samples exposed for 48 h in Figure 4. The sum of Fe and Cr ion concentrations was used, as these two metals are the primary components of SS410, SS430 and SS430F samples. At 80 °C exposure, both SS410 and 430 coupons exhibited high (Fe + Cr) concentrations (>4000 wppm) in the bio-oil, while SS430F had a lower concentration (<300 wppm), indicative of higher corrosion resistance of SS430F over the others. This result agrees with the mass change or loss values of SS410, SS430 and SS430F coupons at 80 °C exposure presented in Figure 2. (Fe + Cr) concentrations from the alloy samples at 35 °C exposure were similar, implying that the SS430 mesh suffered metal dissolution as SS410 samples did. This behavior is also in agreement with the mass change results in Figure 2. At 50 °C exposure, (Fe + Cr) concentrations were lower in SS430 and SS430F coupons than in SS410 samples and SS430 mesh. Overall, SS430F is expected to be most resistant in FR3 bio-oil exposure, while SS430 may corrode as severely as SS410 in the bio-oil at 50 and 80 °C (see SS430 mesh and coupon data in Figure 4).

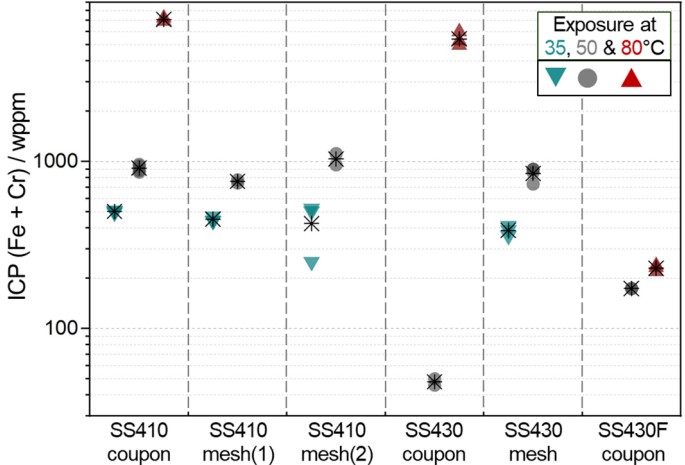

**Figure 4.** ICP-MS based (average Fe + average Cr) concentrations of leached SS410, SS430 and SS430F samples. Downward triangles, circles and upward triangles are individual data at 35, 50 and 80 °C exposures, respectively. Asterisk symbols are the representative mean value of (average Fe + average Cr) concentration data. The raw ICP data for the average Fe and Cr are summarized in Table A2.

Mass change results from exposure tests for 96 and 168 h at 80 °C are summarized in Table 4. SS430F and Ni200 coupons experienced only minor mass changes, implying their high resistance in FR3 bio-oil for extended exposure times at 80 °C. On the other hand, both SS410 and SS430 coupons showed significant mass loss associated with severe corrosion attack. It is repeatedly noted that SS430F coupons were more corrosion resistant than SS430 coupons, regardless of their similar chemical compositions (except S contents). One possibility is that a high sulfur content in SS430F may have mitigated metal dissolution. This assumption is not reviewed here but could be further investigated in a following work.

**Table 4.** Mass change data of SS410, SS430, SS430F and Ni200 coupons in mg·cm$^{-2}$ after exposures at 80 °C for 96 and 168 h, respectively. The numbers without brackets are individual data while the values in brackets are the averages. Two outlier data (not in the table below) are $-0.01$ mg·cm$^{-2}$ from SS430 and $-36.3$ mg·cm$^{-2}$ from SS430F after exposures at 80 °C for 168 h.

|  | **SS410** | **SS430** | **SS430F** | **Ni200** |
|---|---|---|---|---|
| 96 h | - | $-49.4$, $-48.7$ [$-49$] | $+0.002$, $-0.015$ [$-0.007$] | - |
| 168 h | $-113$ | $-47.6$, $-50.4$, $-45$ [$-47.7$] | $0.037$, $0.088$, $-0.004$ [$+0.04$] | $+0.04$, $-0.1$ [$-0.036$] |

To investigate any preferential metal dissolution, ICP-MS data were used to calculate Fe/Cr ratios for specific bio-oil exposure cases, and the results are summarized in Tables 5–7. For SS410 coupons, preferential Fe dissolution, as noted by Fe/Cr greater than the alloy Fe/Cr, was more significant for 35 °C exposure than 50 and 80 °C exposures (see Table 5). Considering the lowest mass loss after 35 °C exposure, it is presumed that the greater dissolution of Fe than Cr should be connected to corrosion mitigation. One possibility is that the alloy surface became Cr enriched due to more Fe removal, and formed a protective passive film to delay further dissolution.

**Table 5.** Fe/Cr values from ICP-MS data for SS410 coupons after exposures at 35, 50 and 80 °C for 48 h. The raw ICP data for Fe and Cr are summarized in Table A2.

|  | **Fe/Cr of Alloy (from Table 1)** | **Fe/Cr ICP (35 °C Exposure)** | **Fe/Cr ICP (50 °C Exposure)** | **Fe/Cr ICP (80 °C Exposure)** |
|---|---|---|---|---|
| Individual data | 7.4 | 13.1 13.4 13.6 | 8.2 8.7 9 9.2 | 8.8 9.1 |
| Note | - | More Fe dissolution | Slightly more Fe dissolution | Slightly more Fe dissolution |
| Avg. mass change in Figure 2 | - | $-1.7$ mg·cm$^{-2}$ | $-5.7$ mg·cm$^{-2}$ | $-43$ mg·cm$^{-2}$ |

**Table 6.** Fe/Cr values from ICP-MS data for SS430 and SS430F coupons as well as SS430 mesh after exposures at 50 °C for 48 h. The raw ICP data for Fe and Cr are in Table A2.

|  | SS430F Coupon | SS430 Coupon | SS430 Mesh |
|---|---|---|---|
| Fe/Cr of alloy (From Table 1) | 4.6 | 4.8 | 4.7 |
| Fe/Cr ICP | 15.8 17.9 | 15 18.8 | 5.4 5.9 6 |
| Note | More Fe dissolution | More Fe dissolution | Slightly more Fe dissolution |
| Avg. mass change in Figure 2 | 0 mg·cm$^{-2}$ | 0 mg·cm$^{-2}$ | −8.5 mg·cm$^{-2}$ |

**Table 7.** Fe/Cr values from ICP-MS data for SS430 and SS430F coupons after exposures at 80 °C for 48 h. The raw ICP data for Fe and Cr are in Table A2.

|  | SS430F Coupon | SS430 Coupon |
|---|---|---|
| Fe/Cr of alloy (From Table 1) | 4.6 | 4.8 |
| Fe/Cr ICP | 17.1 18.9 21.1 | 4.5 5.3 |
| Note | More Fe dissolution | Congruent dissolution of Fe and Cr |
| Avg. mass change in Figure 2 | 0 mg·cm$^{-2}$ | −37 mg·cm$^{-2}$ |

Fe/Cr values at 50 °C exposure in the bio-oil are compared for SS430 coupons and meshes as well as SS430F coupons in Table 6. Both SS430 and SS430F coupons, which showed almost no mass change, had large Fe/Cr values (~15 or greater) compared to Fe/Cr of the alloys, indicating that Fe was removed preferentially over Cr. Based on this, the alloys must have had Cr-rich surfaces associated with their high corrosion resistance at 50 °C bio-oil exposure. On the other hand, Fe/Cr values of SS430 meshes were very similar to Fe/Cr of the alloy substrate, suggesting congruent dissolution of Fe and Cr without any possible Cr enrichment at the surfaces. Considering the relatively high mass loss (−8.5 mg·cm$^{-2}$) of the SS430 meshes, it is reasonable to assume that absence of Cr enrichment (as evidenced by congruent dissolution) at the surface caused uninterrupted corrosion of the alloy during bio-oil exposure.

Fe/Cr values at 80 °C exposure are compared for SS430 and SS430F coupons in Table 7. Fe/Cr values of SS430F and SS430 indicated their Fe preferential and congruent dissolution behaviors, respectively. Similar to the trends confirmed in Table 6, larger Fe/Cr values, implying surface Cr enrichment, were associated with insignificant mass change, thereby corrosion resistance of the alloy. Apparently, comparison of Fe/Cr values from ICP-MS data were useful to distinguish congruent and preferential Fe dissolution behaviors that were correlated with the corrosion mass changes. To further investigate corrosion attack and any feature leading to Cr enrichment, cross-sectional SEM/EDS characterization was performed for selected post-exposure alloy samples.

### 3.2. Post Exposure Sample Characterization

A cross-sectional SEM image and its EDS maps are presented for the SS410 coupon after exposure at 50 °C for 48 h in Figure 5. Cr enriched spots were identified in both the surface and substrate while oxygen was enriched in the surface area. It appeared that no continuous Cr-rich oxide layer was formed on the alloy surface, which would be related

to low corrosion resistance. Indeed, this SS410 coupon showed a relatively high mass loss, $-6.3$ mg·cm$^{-2}$. The Cr enriched spots in the substrate are presumably Cr carbides considering a high carbon content of the alloy (0.13 wt.%, see Table 1). The formation of Cr carbide would reduce the amount of Cr available to form a passive oxide layer, and could lower the corrosion resistance of SS410.

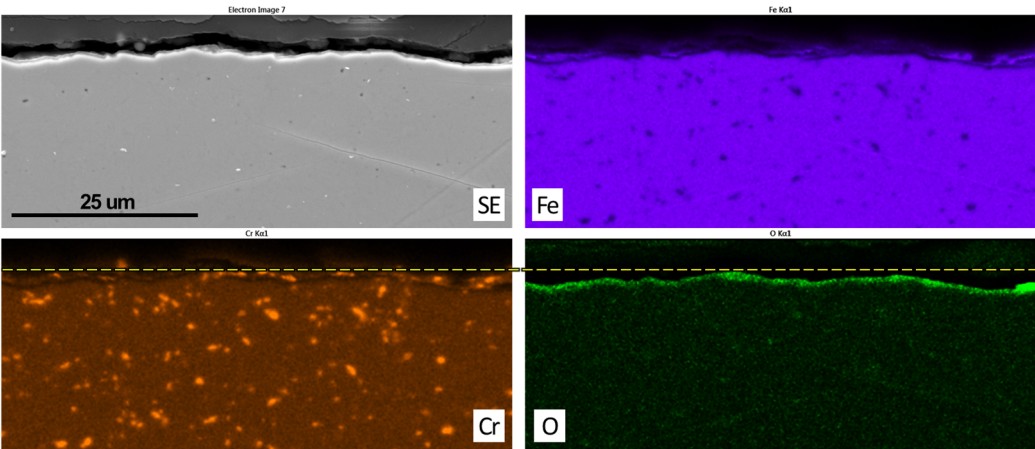

**Figure 5.** A secondary electron image and EDS maps from a cross-section location of SS410 coupon after exposure at 50 °C for 48 h. The mass change of the sample was $-6.3$ mg·cm$^{-2}$. The dashed line for Cr and O maps is used to visually guide a surface level.

As an example of a low mass change sample ($-0.026$ mg·cm$^{-2}$), SS430F coupon after exposure at 80 °C for 48 h was prepared for microscopic analysis and cross-sectionally characterized using SEM and EDS as shown in Figure 6. Despite low resolution for Fe and Cr signals in EDS maps, it is observed that Cr, Fe and O were uniformly distributed in the surface layer without a significant change in concentration of metal elements. This could indicate that the surface layer was composed of compact and homogenous Fe and Cr oxides, and it protected the alloy from corrosion attack. Elongated Mn and Cr sulfide phases, which did not overlap, were also observed in the substrate of SS430F due to a high sulfur content of the alloy.

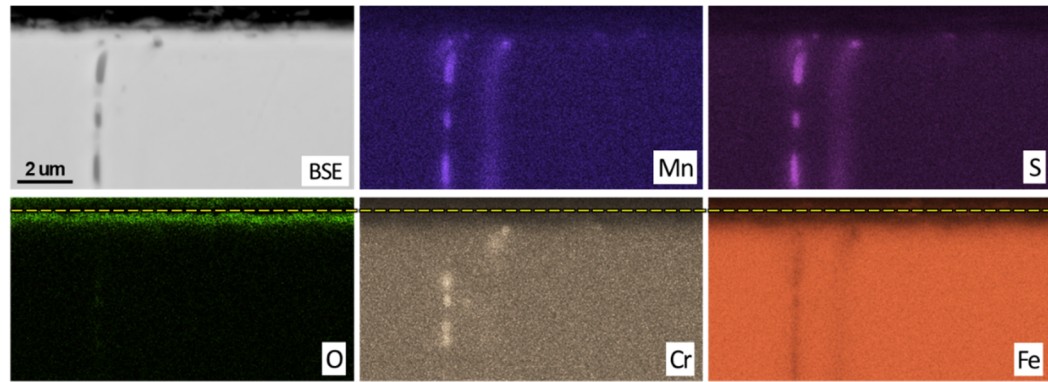

**Figure 6.** A back scattered electron (BSE) image and EDS maps from a cross-section location of SS430F coupon after exposure at 80 °C for 48 h. The mass change of the sample was $-0.026$ mg·cm$^{-2}$. The dashed line for Cr, Fe and O maps is used to visually guide a surface level.

SS430 coupon after exposure at 80 °C for 48 h, as an example of a high mass change sample ($-38$ mg·cm$^{-2}$), was characterized using SEM and EDS as presented in Figure 7. Unlike the samples where surface oxides were present (see Figures 5 and 6), this post-exposure sample did not exhibit an O-rich surface layer. This observation suggests that the characterized SS430 coupon did not form a protective surface oxide layer so that metal

dissolution was not prevented during bio-oil exposure. Regarding the type of corrosion, localized or preferential attack was not observed in the post-exposure SS430 coupon, indicating congruent dissolution of metal constituents. Note that congruent dissolution of the SS430 coupon was also inferred by ICP-MS data analysis presented in Table 7.

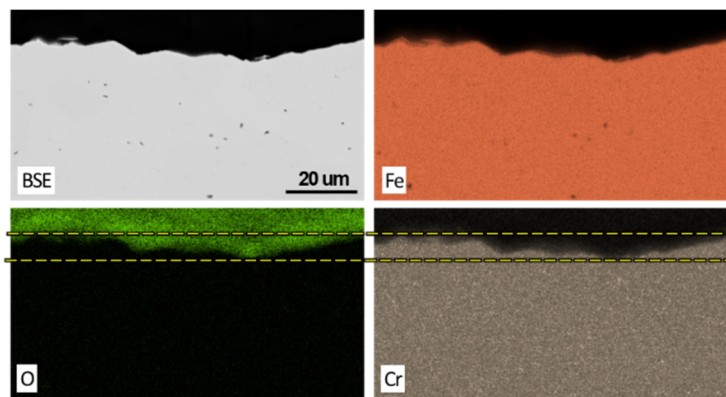

**Figure 7.** A BSE image and EDS maps from a cross-section location of SS430 coupon after exposure at 80 °C for 48 h. The mass change of the sample was $-38$ mg·cm$^{-2}$. The dashed lines for Cr and O maps are used to visually guide two surface levels. Note that the oxygen signal was from an epoxy mount.

Besides the post-exposure alloy coupons, SS410(1) and SS430 meshes after exposure at 50 °C for 48 h were characterized cross-sectionally as presented in Figures 8 and 9. Apparent corrosion attack, characterized by significant pitting (See Figure 8a,b), was observed in SS410(1) mesh, which had a ~8 mg·cm$^{-2}$ mass loss. On the corroded surfaces, some precipitates with relatively dark contrast were also visible (see the arrows in Figure 8b,c). In EDS maps, the precipitates appeared to be Cr-rich nodules with O content which seem similar to the discontinuous Cr-rich oxide layer observed in the SS410 coupon in Figure 5. As no corrosion protection is expected from such a discontinuous oxide, it is reasonable to observe severe corrosion attack in the SS410 mesh.

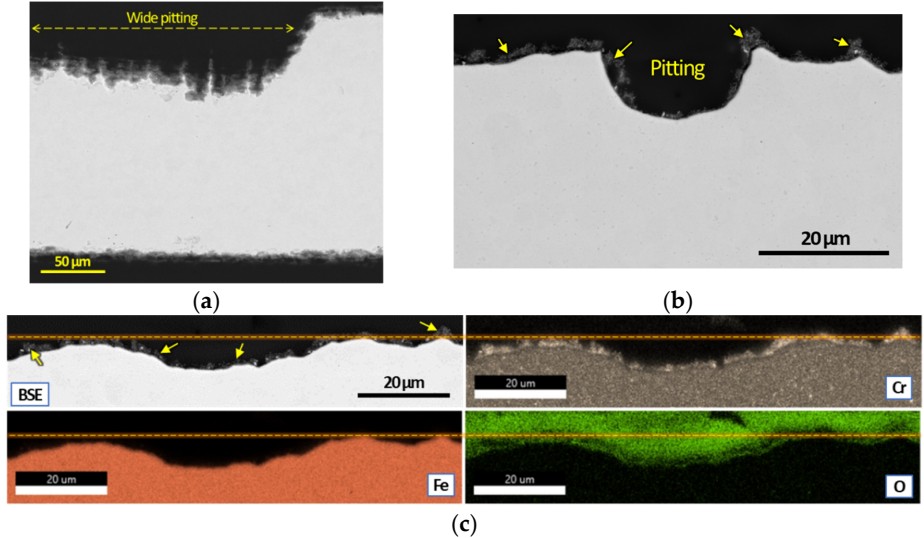

**Figure 8.** Cross-sectional SEM and EDS characterization of SS410 mesh (1) after exposure at 50 °C for 48 h; (**a,b**) pitting attacks and (**c**) BSE image and EDS maps of another location. For (**b,c**), small arrows are used to guide to locations of surface precipitates. The mass change of the sample was $-8$ mg·cm$^{-2}$. The dashed lines in (**c**) are used to visually guide a surface level. Note that the oxygen signal was from an epoxy mount.

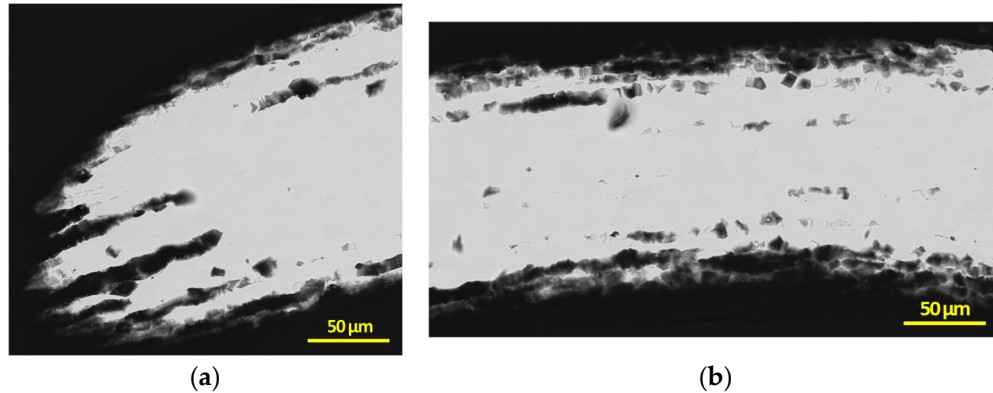

(**a**)  (**b**)

**Figure 9.** Cross-sectional BSE images of SS430 mesh after exposure at 50 °C for 48 h; (**a**,**b**) severe corrosion attacks in two different locations. The mass change of the sample was −8.6 mg·cm$^{-2}$.

Two different locations of a post-exposure SS430 mesh are presented in cross-sectional view SEM images in Figure 9. As clearly seen from the images, severe corrosion attack is evident in the alloy surface and even within the substrate. Based on ICP-MS Fe/Cr values summarized in Table 6, SS430 mesh exposed in bio-oil at 50 °C must have had congruent metal dissolution without any Cr enrichment for forming protective surface oxide. As a result of uninterrupted metal dissolution, SS430 mesh should present notable corrosion attack, and indeed, severe corrosion attack was observed in post-exposure SS430 mesh shown in Figure 9.

### 3.3. Comprehensive Data Analysis

The mass change results presented in Figures 2 and 3, as well as Table 4, are useful to identify corrosion-resistant alloys in FR3 bio-oil. Among the tested alloys, SS430F and Ni200 were highly resistant to corrosion even in a severe exposure condition at 80 °C for 168 h. Meanwhile, SS430 coupons and meshes, which had similar Cr content (~17 wt.%) to SS430F, suffered severe corrosion except the cases when SS430 wires were used. Based on these results, the predicted corrosion compatibility of the six alloys are summarized in Table 8.

While Cr is highly crucial to improve alloy corrosion resistance, the content of Cr is not considered as the sole factor for alloy mass change results in FR3 bio-oil. Another finding from mass change data is that Ni, without any Cr alloying, is naturally resistant to bio-oil environments, which predicts excellent corrosion resistance of Ni-based alloys in bio-oil environments but contradicts a previous report [42]. Other factors affecting corrosion resistance, not discussed in this work, could be investigated in another study.

However, the role of Cr was distinctive in the alloy samples with high mass loss and low mass change as indicated by ICP-MS data in Tables 6 and 7. For the alloy samples with low mass change, evidence of Cr enrichment, from large Fe/Cr values in FR3 bio-oil, suggests that the alloy formed a Cr-rich protective oxide layer which minimized metal dissolution. Indeed, a uniform/continuous surface oxide layer containing Cr and Fe was observed in a post-exposure SS430F coupon with minimal mass change (see Figure 6). On the other hand, Fe/Cr values of bio-oil close to the Fe/Cr values of alloy samples imply congruent dissolution of Fe and Cr where no Cr enrichment would occur. Consequently, the alloy samples that experienced congruent metal dissolution would exhibit high mass loss, which agrees with the results in Tables 6 and 7. The conditions where surface Cr enrichment is promoted were not clearly revealed for the alloy samples tested in this work. To further understand Cr enrichment during bio-oil exposure, any possible alloy element or microstructural feature known to affect Cr diffusion should be investigated.

**Table 8.** Predicted corrosion compatibility of five stainless steels and Ni200 in FR3 bio-oil at three exposure temperatures. Color codes are used to visually distinguish the corrosion compatibility.

| Corrosion Risk | Low | FR3 Bio-Oil Exposure Temperature | | |
|---|---|---|---|---|
| | Intermediate | 35 °C | 50 °C | 80 °C |
| | High | | | |
| SS410 | | Mass loss (about $-2$ mg·cm$^{-2}$, see Figure 3) | High mass loss ($<-5$ mg·cm$^{-2}$, see Figure 3 and Table 4) | |
| SS430 | | Mass loss (about $-1.8$ mg·cm$^{-2}$, see Figure 3) | Could exhibit high mass loss (about $-9$ mg·cm$^{-2}$, see Figure 3) | High mass loss (about $-50$ mg·cm$^{-2}$, see Table 4) |
| SS430F | | No data, but high resistance expected | Low mass change (within $\pm0.1$ mg·cm$^{-2}$, see Figure 3 and Table 4) | |
| SS304 | | Low mass change (within $\pm0.4$ mg·cm$^{-2}$, see Figure 3) | | Mass loss (about $-1.6$ mg·cm$^{-2}$, see Figure 3) |
| SS316 | | No data, but high resistance expected | Low mass change (within $\pm0.1$ mg·cm$^{-2}$, see Figure 3) | |
| Ni200 | | Low mass loss ($<-1$ mg·cm$^{-2}$, see Figure 3 and Table 4) | | |

## 4. Summary

Quantitative corrosion data of different stainless steels and Ni200 exposed to a fast pyrolysis bio-oil (FR3 bio-oil) were collected using mass change and ICP-MS measurements. Some post-exposure alloy samples were cross-sectionally characterized using SEM and EDS. The key findings are summarized as below.

1. SS430F, SS316 and Ni200 exhibited the highest corrosion resistance in FR3 bio-oil, even for extended exposure (168 h) at 80 °C. SS304 was also corrosion-compatible but sensitive to exposure temperature in wire form. SS410 samples showed increasing mass loss in bio-oil exposure with increasing temperature and time.
2. Fe/Cr values from ICP-MS data implied that the alloy samples with minimal mass change (associated with high corrosion resistance) formed protective surface oxide from Cr enrichment. In contrast, congruent metal dissolution, where Cr enrichment did not occur, is considered dominant in the alloy samples with large mass losses.
3. SEM and EDS characterization showed the formation of a continuous oxide layer with uniform Cr distribution in a SS430F coupon that had minimal mass change after bio-oil exposure. For the alloy samples with notable mass losses, however, an oxide layer was absent or did not have uniform Cr distribution.

Further studies are necessary to understand the chemical reactions and corrosion mechanism associated with the alloy degradation in bio-oils. Effects of organic acids and other oxygenates in bio-oils need to be investigated, especially with low Cr alloys.

**Author Contributions:** Conceptualization, J.J., D.S. and J.R.K.; Methodology, J.J., D.S. and J.R.K.; Validation, J.J., D.S., J.R.F.III, E.C., M.V.O. and M.D.K.; Experiments, J.J., D.S. and J.E.W.IV; Characterization, Y.-F.S. All authors have read and agreed to the published version of the manuscript.

**Funding:** This research was funded by the U.S. Department of Energy's Bioenergy Technologies Office. This manuscript has been authored by UT-Battelle, LLC, under contract DE-AC05-00OR22725 with the US Department of Energy (DOE).

**Institutional Review Board Statement:** Not applicable.

**Acknowledgments:** Timothy Theiss, Sebastien Dryepondt, Sinchul Yeom and Vlad Lobodin at ORNL provided useful comments for this manuscript. The United States Government retains and the publisher, by accepting the article for publication, acknowledges that the United States Government retains a non-exclusive, paid-up, irrevocable, world-wide license to publish or reproduce the published form of this manuscript, or allow others to do so, for United States Government purposes. The Department of Energy will provide public access to these results of federally sponsored research in accordance with the DOE Public Access Plan (http://energy.gov/downloads/doe-public-access-plan, Access date: 1 November 2022).

**Conflicts of Interest:** The authors declare no conflict of interest.

## Appendix A

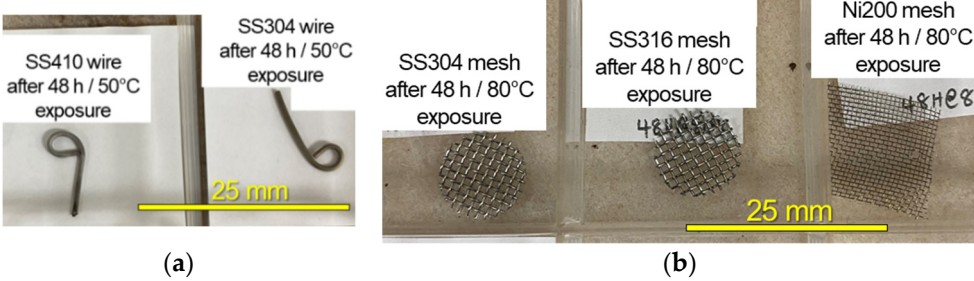

(**a**)  (**b**)

**Figure A1.** Appearance of post-exposure alloy samples in (**a**) wire and (**b**) mesh forms. The alloy types and exposure conditions are labeled for each sample.

**Table A1.** Details of acid digestion step for ICP-MS measurements.

| Step | Description | Duration |
|:---:|:---:|:---:|
| 1 | Microwave-aided Teflon-bomb digestion of acid mixture | - |
| 2 | Stabilization of mixed acids at room temperature | 12~18 h |
| 3 | 0.25 mL bio-oil sample + 1.5 mL 16 M nitric acid (added in 0.1 mL increments), this initial digest was stored with a cover. | - |
| 4 | Stabilization of initial digests | 12~18 h |
| 5 | +0.5 mL 12 M hydrochloric acid + 0.2 mL concentrated hydrofluoric acid | - |
| 6 | Low-level digestion in high pressure microwave rotor (SK15) | - |
| 7 | Three-step heating of the digests to 205 °C | - |
| 8 | Digests diluted to 15 mL | - |

**Table A2.** Raw ICP-MS data containing average and standard deviation (stdev) values for Fe and Cr cations from FR3 bio-oils used for sample exposures at 35, 50 and 80 °C for 48 h.

| Sample | Exposure Temp. in °C | Fe ppm | | Cr ppm | | (A) Fe + Cr ** | Mean Value of (A) ** | Fe/Cr *** |
|---|---|---|---|---|---|---|---|---|
| | | Average | Stdev | Average | Stdev | | | |
| SS410 coupon | 35 | 470 | ±87 | 35 | ±18 | 505 | 503 | 13.6 |
| | 35 | 474 | ±62 | 36 | ±13 | 510 | | 13.1 |
| | 35 | 461 | ±60 | 34 | ±12 | 495 | | 13.4 |
| | 50 * | 795 | ±105 | 91 | ±33 | 886 | 914 | 8.7 |
| | 50 * | 771 | ±101 | 94 | ±34 | 865 | | 8.2 |
| | 50 * | 849 | ±113 | 93 | ±33 | 942 | | 9.2 |
| | 50 * | 867 | ±116 | 96 | ±35 | 963 | | 9 |
| | 80 | 6250 | ±815 | 689 | ±248 | 6939 | 7095 | 9.1 |
| | 80 | 6510 | ±1200 | 740 | ±376 | 7250 | | 8.8 |
| SS410 mesh (1) | 35 | 439 | ±57 | 32 | ±11 | 471 | 452 | 13.9 |
| | 35 | 416 | ±54 | 28 | ±10 | 444 | | 15.1 |
| | 35 | 411 | ±76 | 31 | ±16 | 442 | | 13.4 |
| | 50 | 656 | ±121 | 90 | ±45 | 745 | 761 | 7.4 |
| | 50 | 689 | ±91 | 90 | ±32 | 778 | | 7.8 |
| SS410 mesh (2) | 35 | 484 | ±63 | 39 | ±14 | 523 | 426 | 12.4 |
| | 35 | 468 | ±105 | 36 | ±22 | 504 | | 13.1 |
| | 35 | 240 | ±31 | 12 | ±4 | 252 | | 20.5 |
| | 50 | 845 | ±161 | 111 | ±57 | 956 | 1036 | 7.6 |
| | 50 | 994 | ±131 | 122 | ±44 | 1116 | | 8.1 |
| SS430 coupon | 50 | 44 | ±8 | 2.3 | ±1.2 | 46 | 48 | 18.8 |
| | 50 | 47 | ±9 | 3.1 | ±1.6 | 50 | | 15 |
| | 80 | 4070 | ±935 | 912 | ±570 | 4982 | 5420 | 4.5 |
| | 80 | 4330 | ±578 | 952 | ±345 | 5282 | | 4.5 |
| | 80 | 5050 | ±657 | 946 | ±340 | 5996 | | 5.3 |
| SS430 mesh | 35 | 323 | ±42 | 36 | ±13 | 359 | 387 | 8.9 |
| | 35 | 361 | ±67 | 29 | ±15 | 390 | | 12.7 |
| | 35 | 380 | ±70 | 33 | ±17 | 413 | | 11.6 |
| | 50 | 619 | ±141 | 115 | ±72 | 734 | 849 | 5.4 |
| | 50 | 770 | ±144 | 128 | ±65 | 898 | | 6 |
| | 50 | 738 | ±136 | 124 | ±64 | 862 | | 6 |
| | 50 | 771 | ±143 | 130 | ±66 | 901 | | 5.9 |
| SS430F coupon | 50 | 163 | ±21 | 10 | ±4 | 173 | 174 | 15.8 |
| | 50 | 165 | ±22 | 9 | ±3 | 174 | | 17.9 |
| | 80 | 222 | ±29 | 11 | ±4 | 233 | 230 | 21.1 |
| | 80 | 208 | ±38 | 11 | ±6 | 219 | | 18.9 |
| | 80 | 226 | ±42 | 13 | ±7 | 239 | | 17.1 |

\* Previously reported in [43], \*\* Used in Figure 4, \*\*\* Used in Tables 5–7.

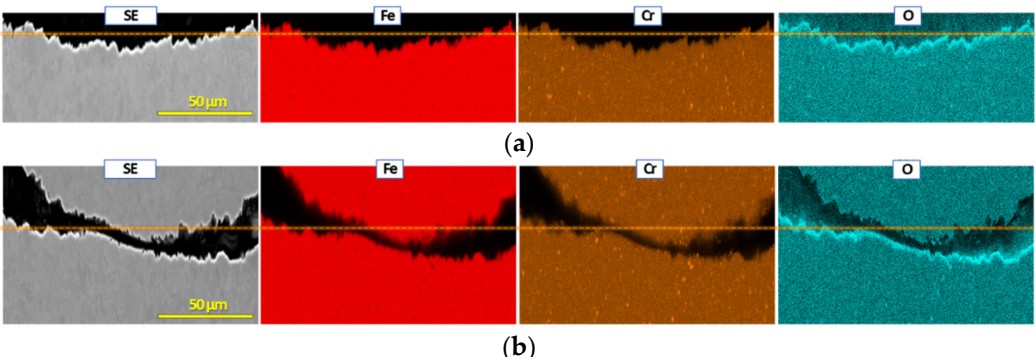

**Figure A2.** Cross-sectional SEM and EDS characterization of SS410 mesh (2) after exposure at 80 °C for 48 h; (**a**,**b**) secondary electron (SE) image and EDS maps from two cross-section locations. The mass change of the sample was $-0.017$ mg·cm$^{-2}$. In both (**a**,**b**), oxygen-rich surfaces with uniform Fe and Cr distribution implies that this mesh sample formed a protective surface oxide that minimized metal dissolution.

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
