# Peer review of "Corrosion Compatibility of Stainless Steels and Nickel in Pyrolysis Biomass-Derived Oil at Elevated Storage Temperatures"

_sustainability, doi:10.3390/su15010022_

Round 1
Reviewer 1 Report
The paper describes corrosion experiments of steel samples exposed to biomass-derived oil at elevated temperatures. The Authors present interesting experimental research. My biggest concern about this paper is that it is rather a technical report, lacking sufficient scientific analysis. My detailed comments are as follows:
1. The literature review is very poor and consists of many auto-citations. Please elaborate briefly on the main results of similar research obtained by other researchers and how do they differ from your research. Why is your research important and novel?
2. In general, the scientific background of the paper is weak. The discussion should be supported by some literature data.
2. The Table 1 caption is too long and contains information, that should be rather placed in the text. The same applies to other tables and figures. Please consider moving some information from the captions into text and footnotes.
3. Lines 111-113: "Alloy samples for characterization were first mounted in epoxy and polished using SiC papers and diamond suspension solutions for a mirror polish surface." - I assume that the cross-sections were made before polishing, and the surfaces of these cross-sections were polished? Please specify.
4. Figures 2, 3 and 4 are hard to read and unclear. Please include the legend instead of the description in the caption.
5. Lines 77-80 Please include the key characteristics of the bio-oil. The Authors can send the readers to previous publications for more detailed information, but the basic characteristics should be included in this paper.
6. Lines 94-96 Please specify the type/composition of the rust remover. Why the time of cleaning was not unified for all samples ("5 min or longer")?
7. The formatting of the paper is not in line with the Instruction for Authors (formatting of tables, figures, subsections etc.)
Reviewer 2 Report
This work presents on the alloy corrosion behavior in a pyrolysis bio-oil, stainless steels with various amount of Cr and Ni and Ni200 were exposed to forest residues (FR) bio-oil at several different temperatures (35, 50 and 80 °C) and times (48, 96 and 168 h). This is an interesting piece of work and suitable to the theme of this journal. There are several comments required to be addressed by the authors of this paper before it could be accepted. The comments are as follows:
1. Page 2, Lines 60-61: Please state the number of metallic alloys in wrought and plate involved in this research as per in this sentence 'Several metallic alloys in wrought rod and plate as well as extruded wire and mesh forms were purchased from different commercial vendors, and their chemical compositions, dimensions and densities are summarized in Tables 1 and 2.'
2. Page 10, Line 265: It is suggested to compare your findings in Table 4 with literature.
3. Section 4 Summary: Please provide future recommendations of your research work.
Reviewer 3 Report
Please revise manuscript follow the comments as shown in attach file.

Round 2
Reviewer 1 Report
The Authors have made improvements in their manuscript. It looks much better now and can be recommended for publication.